# Effect of Whole-Body Cryotherapy on Morphological, Rheological and Biochemical Indices of Blood in People with Multiple Sclerosis

**DOI:** 10.3390/jcm10132833

**Published:** 2021-06-27

**Authors:** Bartłomiej Ptaszek, Aneta Teległów, Justyna Adamiak, Jacek Głodzik, Szymon Podsiadło, Dawid Mucha, Jakub Marchewka, Tomasz Halski, Dariusz Mucha

**Affiliations:** 1Institute of Applied Sciences, University of Physical Education in Krakow, 31-571 Krakow, Poland; justyna.adamiak@awf.krakow.pl (J.A.); jacek.glodzik@awf.krakow.pl (J.G.); 2Institute of Clinical Rehabilitation, University of Physical Education in Krakow, 31-571 Krakow, Poland; aneta.teleglow@awf.krakow.pl (A.T.); szymon.podsiadlo@awf.krakow.pl (S.P.); jakub.marchewka@awf.krakow.pl (J.M.); 3Institute of Health Sciences, Podhale State College of Applied Science in Nowy Targ, 34-400 Nowy Targ, Poland; dawid.mucha@ppuz.edu.pl; 4Faculty of Health Sciences, University of Opole, 45-060 Opole, Poland; tomhalski@wp.pl; 5Institute of Biomedical Sciences, University of Physical Education in Krakow, 31-571 Krakow, Poland; dariusz.mucha@awf.krakow.pl

**Keywords:** whole-body cryotherapy, multiple sclerosis, blood rheology, blood morphology

## Abstract

The aim of this study was to examine and assess the impact of a series of 20 whole-body cryotherapy (WBC) treatments on the biochemical and rheological indices of blood in people with multiple sclerosis. In this prospective controlled study, the experimental group consisted of 15 women aged 34–55 (mean age, 41.53 ± 6.98 years) with diagnosed multiple sclerosis who underwent a series of whole-body cryotherapy treatments. The first control group consisted of 20 women with diagnosed multiple sclerosis. This group had no intervention in the form of whole-body cryotherapy. The second control group consisted of 15 women aged 30–49 years (mean age, 38.47 ± 6.0 years) without neurological diseases and other chronic diseases who also underwent the whole-body cryotherapy treatment. For the analysis of the blood indices, venous blood was taken twice (first, on the day of initiation of whole-body cryotherapy treatments and, second, after a series of 20 cryotherapy treatments). The blood counts were determined using an ABX MICROS 60 hematological analyzer (USA). The LORCA analyzer (Laser–Optical Rotational Cell Analyzer, RR Mechatronics, the Netherlands) was used to study the aggregation and deformability of erythrocytes. The total protein serum measurement was performed using a Cobas 6000 analyzer, Roche and a Proteinogram-Minicap Sebia analyzer. Fibrinogen determinations were made using a Bio-Ksel, Chrom-7 camera. Statistically significant differences and changes after WBC in the levels of red blood cells (RBC), hemoglobin (HGB), hematocrit (HCT), elongation index, total extend of aggregation (AMP), and proteins (including fibrinogen) were observed. However, there was no significant effect of a series of 20 WBC treatments on changes in blood counts, rheology, and biochemistry in women with multiple sclerosis. Our results show that the use of WBC has a positive effect on the rheological properties of the blood of healthy women.

## 1. Introduction

Multiple sclerosis (MS) is a chronic disease of the central nervous system, the key feature of which is the presence of demyelination foci (especially in the white matter of the brain). The pathogenesis of MS is not fully understood. The disease is perceived as an inflammatory demyelinating disease in which damage to the axons plays an important role. At present, there are no current and accurate statistical data assessing the number of people affected by this disease. It is estimated that over 2,500,000 people worldwide suffer from MS, including over 630,000 in Europe and approximately 50,000 in Poland [1,2]. The first symptoms of MS appear mainly in people between 20 and 40 years of age [3].

There is currently a thesis that MS is associated with a combination of hereditary tendencies and undefined environmental factors [4]. Environmental factors that may be indirectly related to the onset of the disease have been investigated. The thesis concerning the significance of environmental factors on the increased risk of morbidity is supported by the fact that the frequency of MS occurrence in different areas differs. In studies of the links between MS and the environment, attention has been paid to the impact of, inter alia, geoclimatic and seasonal, economic and social, and racial and ethnic factors [5].

Environmental factors have been shown to be equally important in controlling MS prevalence and distribution, as well as some aspects of its clinical course and disability progression [6,7]. Particulate matter pollution is one of many environmental factors associated with MS risk and MS relapse [8]. Other MS environmental associated factors include smoking, passive smoking, prenatal smoking, deficient sun-exposure, vitamin-D deficiency, viral infections, obesity, poor lifestyle and dietary habits, and gut microbiota [9,10].

The beneficial effects of cold on the human body have been known for a long time. In Poland, cold therapy has been successfully used for over 30 years [11]. The main reaction that initiates the adaptation process of people in cryochambers is the reduction of blood flow through the skin, which in turn is associated with an increase in the thermal insulation capacity of the skin [12,13]. Many studies (in both healthy and young and middle-aged people) have shown a relationship between the exposure of the whole body to extremely low temperatures and changes in the levels of selected enzymes and hormones in bodily fluids. Morphological and biochemical tests carried out after the application of cryotherapy have shown an increase in blood serum levels of adrenaline, noradrenaline, adrenocorticotropic hormone (ACTH), cortisol, and testosterone (in men) and a decrease in the parameters of inflammatory reactions such as the erythrocyte sedimentation rate (ESR), Waaler–Rose test, and seromucoid [14,15,16,17,18]. After several days of stimulation with cryogenic temperatures, patients with rheumatoid arthritis (RA) and physically active people showed an increase in their HGB and PLT levels, increased creatinine levels, and increased glycemia [19,20]. Some reports have shown a decrease in RBC in trainees [16,21,22,23] and an increase in WBC count [24,25], while others describe no change in RBC and/or WBC, most likely due to too few sessions [16,20,22,23,26]. Blatteis (1998) observed a decrease in WBC and RBC in healthy subjects after a series of treatments [27]. Banfi et al. (2009) demonstrated a decrease in HGB levels with a simultaneous increase in MCH and MCHC values after the end of the procedures [16]. However, there is still a lack of studies assessing the effect of whole-body cryotherapy on morphological indices in patients with MS.

The aim of this study was to investigate and evaluate the impact of a series of 20 whole-body cryotherapy treatments on the biochemical and rheological blood indices of patients with multiple sclerosis.

## 2. Materials and Methods

### 2.1. Materials

This prospective controlled study with 3 arms was carried out at the Bronisław Czech University of Physical Education in Krakow from November 2018 to September 2019. The study was approved by the Bioethical Commission of the Regional Medical Chamber in Krakow (number: 87/KBL/OIL/2018 of 8 May 2018). The studies were also registered on the ANZCTR platform (Australian New Zealand Clinical Trials Registry): ACTRN12620001142921.

#### 2.1.1. Inclusion Criteria

Diagnosis of MS (ICD10:G35) according to the McDonald review criteria (groups with MS) [28];Female;Aged 30–55 years old;Expanded Disability Status Scale (EDSS) with a value from 0 to 6.5 (groups with MS);Written consent of the patient to participate in the study

#### 2.1.2. Exclusion Criteria

Vitamin D supplementation;Consuming more than 4 cups of coffee or more than 2 alcoholic beverages per day;Changing the diet immediately before or during the project;Participation in other forms of physical activity directly before or during the studyContraindications to whole-body cryotherapy (i.e., pregnancy, severe hypertension (BP > 180/100), acute or recent myocardial infarction, unstable angina pectoris, arrhythmia, symptomatic cardiovascular disease, cardiac pacemakers, peripheral arterial occlusive disease, venous thrombosis, acute or recent cerebrovascular accident, uncontrolled seizures, Raynaud’s Syndrome, fever, tumor disease, symptomatic lung disorders, bleeding disorders, severe anemia, infection, cold allergy, and acute kidney and urinary tract diseases)

Eighty people applied, and fifty women were finally selected to participate in the research program. Before joining the study, each of the volunteers read the patient information and, in case of doubts, could ask questions. Then, they gave their written consent to participate in the study (Figure 1).

#### 2.1.3. Characteristics of the Participants

The experimental group (CRYO-MS) consisted of 15 women aged 34–55 (mean age, 41.53 ± 6.98 years) with diagnosed multiple sclerosis who underwent a series of whole-body cryotherapy treatments.

The control group 1 (CONTROL-MS) consisted of 20 women aged 32–48 (mean age, 40.45 ± 4.77) with diagnosed multiple sclerosis. The subjects from this group did not participate in intervention in the form of whole-body cryotherapy treatments. Purposeful sampling was used. The main reason for qualifying the CONTROL-MS group was an organizational inability to participate in WBC procedures (in comparison with the baseline).

The control group 2 (CONTROL-CRYO) consisted of 15 women aged 30–49 years (mean age, 38.47 ± 6.0 years), without neurological diseases and other chronic diseases, who used whole-body cryotherapy (in order to compare the changes in blood parameters between healthy controls and people with MS).

The women were included in the research program after obtaining their consent from a neurologist, physician and, after consultation, with a physiotherapist (for the assessment of their health status, functional status, the course of the disease, and the type of the current treatment and medicines; patients with MS were treated mainly with immunomodulating drugs and steroids). Body weight was determined with the TANITA BC 418 MA device (measurement error, 0.1 kg), and body height was measured with a tape measure (measurement error, 0.5–1 cm).

The characteristics of the Participants are presented in Table 1.

### 2.2. Methods

#### 2.2.1. Description of the Intervention

Whole body cryotherapy (WBC) was carried at the Malopolska Cryotherapy Rehabilitation Center in Krakow. Parameters in the cryochamber were as follows:Atrium temperature: −60 °C (adaptation to low temperature);Chamber temperature: −120 °C.The time of a single WBC session during study period was respectively:1.5 min (1 treatment);2 min (2 treatments);3 min (3–20 treatments).

One treatment a day was performed (every day in the same time period, 3:00 p.m.–5:00 p.m.). There were 20 treatments in total, and they were performed 5 times a week.

Women entering the cryochamber were dressed in swimsuits, high woolen socks, and shoes with a high wooden sole. The parts of the body most exposed to low temperatures were covered (e.g., the subjects had gloves, head covering, and a face mask). After being exposed to cryotherapy, the patients warmed up on a Kettler Corsa cycloergometer without any resistance for 15 min. Treatments were supervised directly by a trained physiotherapist. A doctor was present at the Rehabilitation Center throughout the entire duration of the treatment.

During the procedure, visual contact with the patients was maintained through the thermal windows. The procedure was additionally monitored by a camera, and the patients were given the current treatment time every 30 s. Inside the chamber, there was a visible alarm button, and the cryochamber was equipped with electronic systems monitoring the oxygen content, humidity, and temperature of both the atrium and the proper chamber.

The treatments were performed in a Wroclaw-type cryochamber wherein liquid nitrogen was the refrigerant cooling (Figure 2).

#### 2.2.2. Blood Indices Analysis

For the analysis of blood indices, venous blood was collected twice: at baseline (i.e., on the day the WBC treatments began (study 1)) and after a series of 20 WBC treatments (study 2). For patients who did not undergo the procedures, blood was only collected once. Throughout the project, blood pressure was monitored in the patients undergoing cryotherapy (before and after each WBC treatment). Blood was collected from the subjects on an empty stomach in the morning from the antecubital, cephalic, or median vein into test tubes:With EDTA (for hematological analysis of the whole blood, K2 potassium edetate salt (6 ml) was used as an anticoagulant);With clotting activator (for serum testing, the main activator ingredient was SiO_2_ (6 mL)).

The blood was collected by a qualified laboratory diagnostician, under the supervision of a physician, in accordance with the applicable standards at the Blood Physiology Laboratory of the University of Physical Education in Krakow. The assessment of blood indices were performed in the Laboratory of Blood Physiology and the Laboratory of Skin Physiology of the University of Physical Education in Krakow and in the Department of Analytical and Clinical Biochemistry of the Krakow Branch Oncology Center at the Institute of Maria Skłodowskiej-Curie.

Assessment of hematological parameters of the blood was done using the ABX MICROS 60 hematology analyzer (USA).
WBC (10^9^/L)—White Blood CellsRBC (10^12^/L)—Red Blood CellHGB (g/dl)—HemoglobinHCT (%)—HematocritPLT (10^9^/L)—Platelet CountMCH (pg)—Mean Corpuscular HemoglobinMCV (fl)—Mean Corpuscular VolumeMCHC (g/dl)—Mean Corpuscular Hemoglobin Concentration

##### Assessment of Elongation and Aggregation Indexes

The LORCA analyzer (Laser–Optical Rotational Cell Analyzer, RR Mechatronics, the Netherlands) was used to study the aggregation and deformability of erythrocytes, and the results were given as the elongation and aggregation index (EI and AI). Tests in the aforementioned device were performed within 30 min of blood collection, at 37 °C, according to a standard protocol [29,30,31].

The blood used for determining the elongation index was taken from test tubes in the amount of 25 µL to 5 mL of 0.14 mM PVP (Polyvinylpyrrolidone), dissolved in a buffered saline solution (PBS). The test sample was placed in the measuring chamber between two concentric cylinders which were set in rotation. The laser light, passing through the thin layer of red blood cells suspended in the PBS solution, was deflected, giving a diffraction image on the projection screen. The EI results are given in the range from 0.30 to 60.30 shear stress measured in Pascals (SS). EI is a measure of the amount of deformation of red blood cells as they move through the measurement chamber [29,30,31].

The blood sample, prior to the actual aggregation test, was oxidized by incubation and mixed with carbogen for 15 min. Then, 1.5 mL of blood was introduced into the measuring chamber of the LORCA analyzer. The cylinder was rotated within 120 s with a shear rate of >400 s^−1^. After 10 s, the centrifugation stopped abruptly, and the aggregation of red blood cells began. The result of the computer analysis represents the time dependence of the scattered light intensity (for a specific shear rate), i.e., selectogram [29,30,31,32].

##### Parameters Determining the Kinetics of Erythrocyte Aggregation Were Investigated (Figure 3)

AI (%) (ang. Aggregation Index)AMP (au) (ang. Total Extend of Aggregation)T½ (s) (ang. Half Time Kinetics of Aggregation)

##### Assessment of Protein Level

The total protein in the blood serum was measured using the Cobas 6000, Roche analyzer (following the biuret method):(1)$$\mathrm{protein}+{\;\mathrm{Cu}}^{2+}\;\overset{\mathrm{alkaline}\;\mathrm{enviroment}}{\to }\;\mathrm{Cu}\;\mathrm{complex}-\mathrm{protein}$$

In an alkaline environment, peptide bonds of proteins form a characteristic violet-colored complex with copper ions contained in the biuret reagent. The following reagents were used: R1-sodium hydroxide, 400 mmol/L; potassium sodium tartrate, 89 mmol/L; R1- sodium hydroxide, 400 mmol/L; potassium sodium tartrate, 89 mmol/L; potassium iodide, 61 mmol/L; zinc sulfate, 24.3 mmol/L. Serum was used in all of the samples. The concentrations were automatically calculated by the system using the Proteinogram-Minicap Sebia analyzer capillary method. Marked indicators included:Total protein (g/L)Albumin (g/L)Alfa-1 globulin (g/L)Alfa-2 globulin (g/L)Beta-1 globulin (g/L)Beta-2 globulin (g/L)Gamma globulin (g/L)A/G ratio

##### Determination of Fibrinogen Concentration

For the determination of fibrinogen concentration, 50 µL of plasma was added to 100 µL of thromboplastin with calcium chloride. The basis of the result is the measurement of the time from the addition of the reagent to the clot formation, which is then converted to the unit of measurement g/L. The determinations were made with the Bio-Ksel, Chrom-7 camera-spectrophotometric method.

#### 2.2.3. Statistical Analysis

Data are presented as mean values and standard deviation. The normality of the distributions was verified on the basis of the Shapiro–Wilk test. The differences between the experimental group and the control groups were assessed using the one-way analysis of variance (ANOVA) or, if it was not met, the Kruskal–Wallis test. For post-hoc evaluation, Tukey’s test for unequal sample sizes was used or, respectively, the multiple comparison test of mean rank for all Dunn’s trials. The dependent variables were compared with the student’s t-test for related variables, and in the case of failure to meet its assumptions, with the Wilcoxon test. Independent variables were compared with the Mann–Whitney U test. The significance level of α = 0.05 was adopted in the analyzes. The applied tests verified the two-sided hypotheses. The analyzes were performed using the Statistica 13 package (StatSoft, Dell Inc, Round Rock, TX, USA).

## 3. Results

A statistically significant difference was observed in the baseline RBC levels, which were lower in MS patients (CRYO-MS and CONTROL-MS) (4.23 ± 0.61 10^12^/L and 4.30 ± 0.67 10^12^/L) compared to healthy women (CONTROL-CRYO) (4.80 ± 0.27 10^12^/L). Moreover, healthy women were characterized by a statistically significant (*p* = 0.007) lower RBC level after the application of WBC (4.57 ± 0.28 10^12^/L).

There was also a significantly higher mean baseline level of HGB in the group of healthy women (CONTROL-CRYO) compared to the mean level of patients with MS before WBC (CRYO-MS) (*p* = 0.045). After the use of WBC in the group of healthy women, a statistically significant (*p* = 0.014) lower mean level of HGB (3.81%) was noted in the group of healthy women.

When analyzing the HCT level, a lower baseline level was observed in MS patients receiving treatments (CRYO-MS) (35.97 ± 5.65%) and in the control group (CONTROL-MS) (35.90 ± 5.88%) compared to healthy women (CONTROL-CRYO) (40.54 ± 2.50%). Moreover, healthy women were characterized by a statistically significant (*p* = 0.003) lower HCT level after the application of WBC (38.35 ± 2.20%).

In terms of the EI level, a significant difference was observed in the baseline levels at the shear stress of 4.24–60.30 Pa. They were lower in healthy women (CONTROL-CRYO) compared to women with MS who underwent (CRYO-MS) and do not underwent treatments (CONTROL-MS) (*p* ≤ 0.016). In healthy women, after the use of WBC, an increase in the EI level was noted at SS 2.19–60.30 Pa. The trait variability was on mean at the level of 28.07%.

A statistically significant higher baseline AMP level was observed in healthy women (21.62 ± 4.06 au) compared to the non-WBC control patients (18.16 ± 4.57 au). There were no statistically significant changes in AMP after the use of WBC.

Analyzing the protein levels, while initially lower levels of total protein were observed in the groups of MS patients (70.17 ± 3.98 g/L and 70.29 ± 2.79 g/L) compared to the group of healthy women (73.56 ± 3.68 g/L), there was a statistically significant reduction in the levels in healthy women after applying WBC (71.73 ± 2.53 g/L) (by 2.49%). Mean albumin levels in healthy women after the use of WBC (CONTROL-CRYO) slightly decreased (*p* = 0.009). There were no statistically significant changes in albumin levels after the use of WBC. Initially lower levels of gamma globulin were observed in the groups of MS patients (CRYO-MS and CONTROL-MS) (9.41 ± 2.22 g/L and 9.47 ± 2.29 g/L) compared to healthy patients (CONTROL-CRYO) (11.53 ± 2.37 g/L) on mean by 18.13%. There were no statistically significant changes after the application of WBC.

Analyzing the mean levels of fibrinogen, a statistically significant increase was observed in healthy women after the use of WBC (CONTROL-CRYO) (34.13%). However, this was considered statistically insignificant due to the large discrepancy in patients with MS (CRYO-MS) (44.24%). There were no statistically significant differences in the baseline levels between groups.

Detailed results are presented in Figure 3 and Figure 4 and Table 2, Table 3, Table 4 and Table 5.

## 4. Discussion

During the past decades, a better understanding of relapsing-remitting multiple sclerosis disease mechanisms has led to the development of several disease-modifying therapies, reducing relapse rates and severity through immune system modulation or suppression. In contrast, current therapeutic options for progressive multiple sclerosis remain comparatively disappointing and challenging. One possible explanation for this is a lack of understanding of the pathogenic mechanisms driving progressive multiple sclerosis. [33]. Currently, there are opportunities to reduce disease activity, inhibit its progression, modify the autoimmune process, influence the intensity of relapses, and alleviate and treat concomitant clinical symptoms. MS therapy is a difficult process which is related, inter alia, to the multitude of symptoms and their overlapping [34,35,36]. Despite a more detailed knowledge about the pathomechanism of the disease, the possibilities for pharmacological treatment that may be widely used are quite limited. Therefore, symptomatic treatment, including physical activity and physiotherapy, plays an important role in addition to pharmacotherapy [37,38].

Blood rheology is the science which studies the flow of blood through blood vessels. It concerns both whole blood, plasma, and morphotic elements (especially red blood cells) [39]. The basic factors that influence blood flow are RBC, red blood cell deformability and aggregation, plasma viscosity, and HCT [39,40,41]. The phenomenon of erythrocyte deformability plays an important role in the flow of blood cells through capillaries with a diameter even two times smaller than themselves [42].

According to Maeda (1996), the correct deformability of erythrocytes is a key role in the blood flow in the vascular system. On the other hand, the proper shape of erythrocytes, intracellular viscosity, and stiffness of their cell membrane walls depend on the level of HGB, which has a significant influence on the deformation capacity. The deformability of red blood cells is very important, as it allows them to pass through the capillaries. The lower the deformability, the higher the blood viscosity and the worse the blood flow in the microcirculation [43].

The phenomenon of spontaneous aggregation of red blood cells in whole blood (i.e., the formation of three-dimensional erythrocyte structures) is a reversible physiological phenomenon which plays a significant role in blood flow at low shear rates and significantly increases blood viscosity [31]. The places particularly susceptible to the formation of erythrocyte aggregates are small blood vessels, where the shear rates are usually low. Ultimately, this causes a decrease in blood flow velocity, or even its inhibition, with the consequence that cells and tissues are under-oxygenated [44].

Hematological changes in MS patients are not characteristic. It is difficult to define a homogeneous mechanism of hematological changes for this disease. Such a mechanism may not exist at all. It has been suggested that hematological changes in MS may be related to factors such as complex humoral and cellular reaction, biochemical disorders, changes related to the attachment and transport of vitamin B12, applied pharmacological treatment, and the phases of the disease (i.e., the presence of relapse or remission) [45,46,47].

The elasticity of erythrocytes is significantly influenced, inter alia, by intracellular viscosity, which is affected by HGB. The corresponding values of the MCH and MCHC parameters determine the possibility of deformability for erythrocytes, even in the case of hyper or hypotonic changes in the blood environment [39]. Al-Din et al. (1991) did not find significant differences in the level of HGB in patients with MS compared to the values obtained in patients with other neurological diseases, and studies have confirmed the thesis of Reynolds and Linnell (1987) concerning the tendency in MS patients for macrocytosis without simultaneous anemia. In the conducted studies, a statistically significant higher mean baseline level of HGB was observed in the group of healthy women compared to the mean level of MS patients [47,48]. After using WBC in the group of healthy women, a statistically significant lower mean level of HGB was noted. While the conducted analysis showed no statistically significant changes in MCH after the use of WBC and in the baseline levels, a statistically significant minimal increase in MCHC was observed in healthy women after the use of WBC.

Grasso et al. (1992) found no pathological HGB values or deviation from the RBC norms in MS patients. They performed blood tests of MS patients and found no difference in MCV compared to the control group. At the same time, due to the fact that the comparative group consisted of people who were not completely healthy, the results were also compared with the accepted physiological norms. It was noted that only one person from the experimental group and one person from the control group had MCV results above the accepted physiological norms [49]. In subsequent studies, Kocer et al. (2009) also failed to confirm macrocytosis in MS patients examined during a relapse. In their research, MCV was within the normal range in over 77% of the respondents while microcytosis was observed in others [50]. De Freitas et al. (2010) also found no differences in MCV in both MS patients undergoing steroid therapy, and in MS patients not undergoing pharmacological treatment [51].

In the conducted studies, there were also no statistically significant changes in WBC in MS patients compared to the control group. However, when analyzing RBC, a statistically significant difference was observed in the baseline RBC levels, with lower levels in patients with MS compared to healthy women. The MCV initially also did not differ between groups, and only a slight reduction in MCV was observed in healthy women after the use of WBC.

Morphological changes in red blood cells (macrocytes and echinocytes) were positively correlated with the severity of MS disease and may impair red blood cell deformity [52].

Platelets also play an important role in the coagulation cascade and are abundant in MS lesions [53]. Platelets themselves do not directly affect erythrocyte aggregation, but affect thrombotic processes [54,55], which supports the idea of ischemic tissue damage [56]. In this study, no statistically significant changes in PLT were found after the use of WBC or in the baseline levels.

Brunetti et al. (1981) observed an increased value of the viscosity indices of whole blood with the simultaneous level of plasma viscosity and HCT not deviating from the norms. They suggested that the increase in the level of blood viscosity was due to a decrease in the deformability of the red blood cells [45]. Later studies by Pollock et al. (1982) did not confirm this hypothesis, as they showed no visible differences in the level of red blood cell deformability between MS patients and the control group of healthy people [46]. Research related to the morphology of erythrocytes was also carried out by Simpson et al. (1987), who reported impaired deformability of erythrocytes in MS patients. In women, significantly higher values of blood viscosity were observed compared to the control group at three shear rates (1 s^−1^, 10 s^−1^, and 100 s^−1^) while in men the blood viscosity was statistically higher only at one shear rate (1 s^−1^) [57].

When analyzing the level of HCT in the conducted studies, a statistically significant difference was observed in the baseline levels, with a lower level in MS patients compared to healthy women. Moreover, healthy women were characterized by a statistically significant lower level of HCT after the use of WBC. On the other hand, when analyzing the EI level, a statistically significant difference was observed in the baseline levels for SS 4.24–60.30 Pa. It was unexpectedly lower in healthy women compared to MS women using and not using the procedures. In healthy women after the use of WBC, a favorable increase in EI was noted at SS 2.19–60.30 Pa. The use of WBC significantly increased the deformation capacity of erythrocytes and decreased the HCT value in healthy women, which positively influenced the rheological properties of blood. In this study, the mean AI level in the experimental group and control group did not change significantly before and after whole-body cryotherapy. There were also no statistically significant changes in AI and T½ at baseline levels.

Fibrinogen is glycoprotein synthesized in the liver. It is involved in blood clotting, hemostasis, and also in inflammation and tissue repair [58,59]. Plasma fibrinogen levels rise two to three-fold during the inflammatory response, which causes cell aggregation and increases blood viscosity [60]. Studies have shown that fibrinogen can modulate the inflammatory response by activating leukocytes and synthesizing pro-inflammatory mediators (i.e., cytokines and chemokines) [58,61]. Fibrinogen is not a clear indicator of the disruption of the Blood–Brain Barrier (BBB). However, it does activate glial cells, leading to BBB dysfunction in MS patients [62].

Miranda Acuña et al. (2017) found in their studies that high plasma fibrinogen levels were associated with active changes in Magnetic Resonance Imaging in patients with MS during relapse [63]. An earlier study by Ehling et al. (2011) on fibrinogen in MS patients showed no elevation of fibrinogen in the cerebrospinal fluid (CSF) or plasma, but this probably because they compared fibrinogen levels between MS patients and patients with central nervous system infections. In addition, while less than a third of MS patients had an acute relapse at the time of fibrinogen analysis, the rest of the patients had an inactive disease or chronic progressive course [64].

Despite the inflammatory events, it seems that patients with relapsing-remitting MS do not have elevated levels of fibrinogen in remission [65]. However, D-dimer levels are elevated [66] and low fibrinogen levels during remission (MS patients vs. controls) do not rule out the possibility of increasing fibrinogen levels during relapse, especially considering the role of fibrinogen in MS pathology where fibrin is involved in the release of cytokines and the activation of microglia in the central nervous system [67,68]. Analyzing the mean levels of fibrinogen in our own studies, no increased levels were observed in women with MS, while a statistically significant increase in the level of fibrinogen was noted after the use of a series of 20 WBC treatments in healthy women. However, this was considered statistically insignificant due to the large discrepancy in patients with MS. In clinical practice, plasma fibrinogen may be an important and easy biomarker of activity during relapse in MS patients, but prospective studies in larger groups are needed to confirm the results.

Studies on the effect of WBC in MS patients were also conducted by Bryczkowska et al. (2018). In the case of healthy people, the first changes in the lipid profile were observed after 20 daily WBC treatments [24], while in people with MS after a series of 30 WBC treatments, researchers did not observe significant changes in total protein, albumin, UA and glucose levels and lipid profile [69]. In this study, initially lower levels of total protein were observed in the groups of MS patients compared to healthy women, and a statistically significant reduction in the level in healthy women after using WBC. There were also no significant differences related to other indexes of the proteinogram (i.e., albumin, alpha-1 globulin, alpha-2 globulin, beta-1 globulin, and beta-2 globulin), except for initially lower levels of gamma globulin in the groups of MS patients compared to healthy women.

Studying the role of red blood cells in MS could reveal further specific differences that could be used as disease biomarkers. In addition, research into the role of red blood cells in MS could broaden our understanding of the pathological mechanisms of this complex and heterogeneous disease, and this in turn could lead to the discovery of new and innovative therapeutic targets that could significantly improve patients’ quality of life. In summary, in the light of the available literature and based on the results of our own research, it can be concluded that WBC is an effective method of combating or inhibiting the progress of many diseases and their negative effects, which is an important factor in maintaining the best possible fitness of the body. To the best of our knowledge, the present study is the first to assess the influence of WBC on the rheological properties of blood in MS women, including EI and AI levels. No pathological or harmful changes were observed after the use of WBC.

## 5. Conclusions

There was no significant effect of a series of 20 systemic cryotherapy treatments on changes in blood counts, rheology, and biochemistry in women with multiple sclerosis.The use of whole-body cryotherapy significantly increases the deformation capacity of erythrocytes and reduces the hematocrit value (within physiological norms) of healthy women, which has a positive effect on the rheological properties of blood.WBC is a safe form of therapy in MS patients, as changes in blood rheology are not responsible for the effectiveness of treatment and WBC does not adversely affect red blood cell deformability and aggregation.

### Study Limitation

This study is not without its limitations. An important aspect that could have affected trial results is the relatively small number of people in the experimental groups. The obtained results are important for determining the safety of WBC in the field of clinical trials, and the research should be continued (with randomization) in larger and more diverse groups of patients.

## Figures and Tables

**Figure 1 jcm-10-02833-f001:**
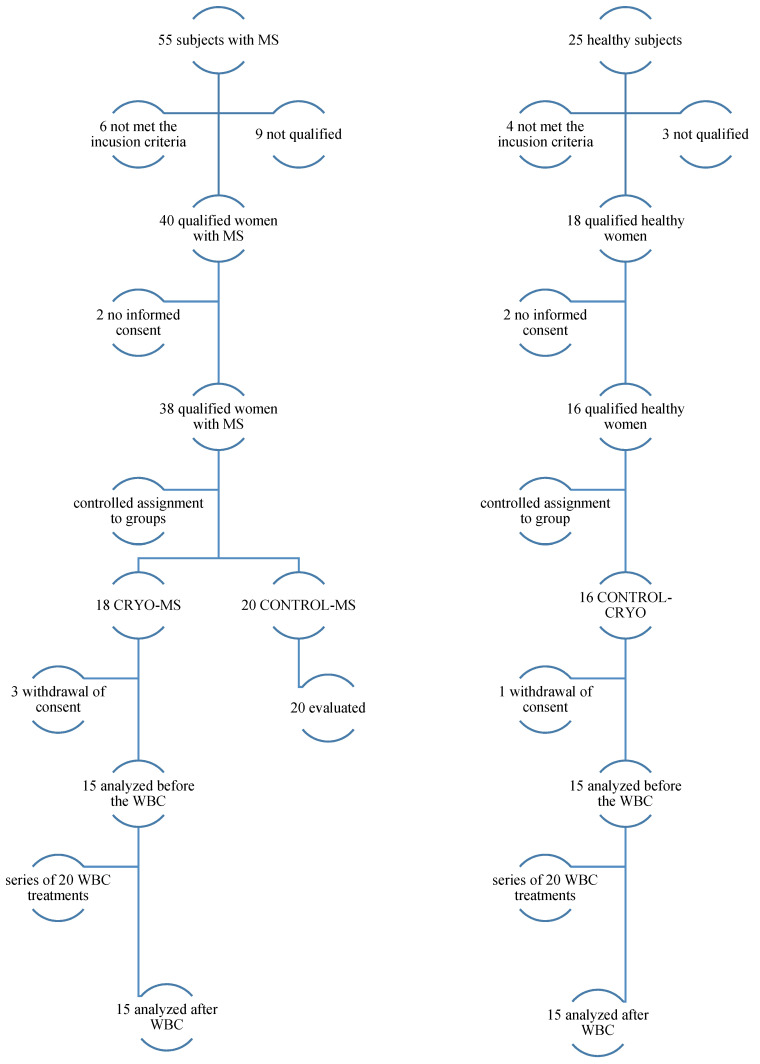
The course of the study.

**Figure 2 jcm-10-02833-f002:**
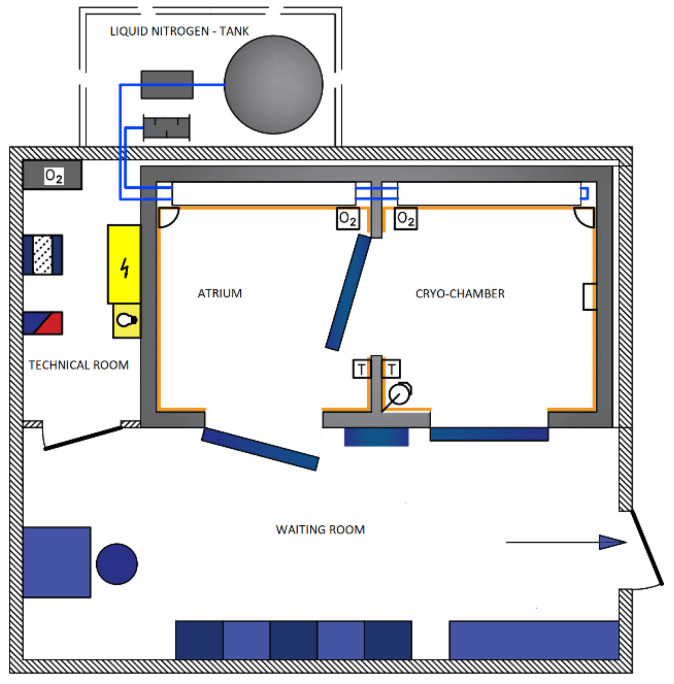
General diagram of a cryochamber (according to CREATOR Sp. z o.o).

**Figure 3 jcm-10-02833-f003:**
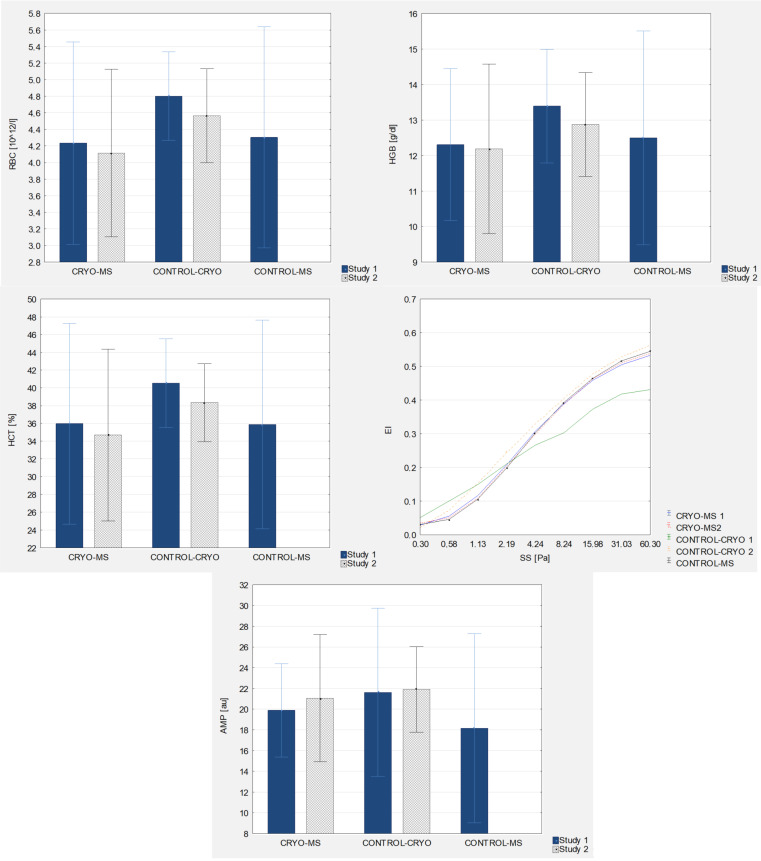
Morphological and rheological indices of blood in experimental group and control groups (study 1 and study 2, respectively).

**Figure 4 jcm-10-02833-f004:**
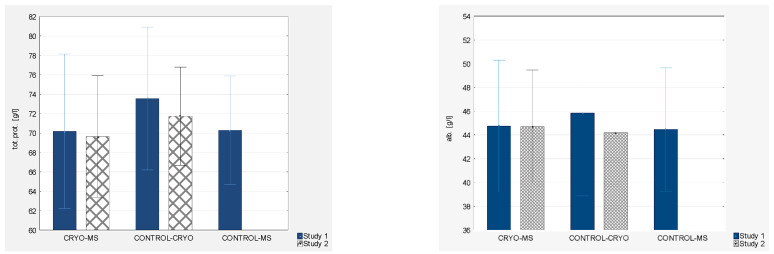
Biochemical indices (total protein, albumin, gamma globulin, fibrinogen) of blood in experimental group and control groups (study 1 and study 2, respectively).

**Table 1 jcm-10-02833-t001:** Characteristics of the control groups and the experimental group.

		Experimental Group CRYO-MS	Control Group 1 CONTROL-MS	Control Group 2 CONTROL-CRYO
Age (years)		41.53 ± 6.98	40.45 ± 4.77	38.47 ± 6.00
Body height (cm)		165.93 ± 6.53	167.25 ± 5.85	169.4 ± 5.79
Body weight (kg)		66.75 ± 16.78	68.78 ± 16.54	72.35 ± 13.85
Body mass index (BMI) (kg/m^2^)		24.18 ± 5.68	24.60 ± 6.04	25.22 ± 4.81
FAT% (%)		33.26 ± 7.45	34.29 ± 8.23	30.47 ± 6.65
FAT (kg)		23.31 ± 11.40	24.81 ± 11.47	22.82 ± 8.99
Fat free mass (FFM) (kg)		43.45 ± 5.68	43.98 ± 5.44	49.55 ± 5.90
Total body water (TBW) (kg)		31.83 ± 4.21	32.23 ± 4.02	36.28 ± 4.32
Expanded Disability Status Scale (EDSS)		3.03 ± 1.67	3.08 ± 1.54	-
Duration of illness (years)		11.00 ± 6.49	13.10 ± 5.45	-
Place of residence (%)	City (100–199 k inhabitants)	20.00	15.00	6.67
	City (200 k inhabitants or more)	73.33	65.00	86.67
	Countryside	6.67	20.00	6.67
Education (%)	Secondary	26.67	5.00	13.33
	Higher	73.33	95.00	86.67
Employment status (%)	Currently employed	86.67	90.00	93.33
	Unemployed	13.33	10.00	6.67
Course of disease (%)	Primary progressive	13.33	-	-
	Relapsing-remitting	86.67	100.00	-
Occurrence of relapses (%)	Several times a year	6.67	5.00	-
	Once a year	20.00	25.00	-
	Every few years	60.00	65.00	-
	No relapse; MS progresses	13.33	5.00	-
Occurring disorders (%)	Spasticity	40.00	40.00	-
	Tremor	6.67	10.00	-
	Excessive fatigue	80.00	90.00	-
	Blurred vision	20.00	35.00	-
	Paresthesia	46.67	30.00	-
	Balance disorders	46.67	55.00	-
	Mood disorders	53.33	35.00	-
	Bladder dysfunction	33.33	35.00	-
Pharmacological treatment (%)	Immunomodulating drugs	67.67	85.00	-
	Steroid drugs	33.33	20.00	-
	None	6.67	5.00	-
	Low Dose Naltrexone (LDN)	6.67	-	-
Psychoactive substances (%)	Coffee	60.00	75.00	67.67
	Cigarettes	6.67	-	-
	Alcohol	-	15.00	6.67
Orthopedic aids (%)	Orthopedic crutches	-	15.00	-
	Nordic walking sticks	13.33	15.00	-

**Table 2 jcm-10-02833-t002:** ANOVA results: baseline parameters (before WBC).

Parameter	CRYO-MS N = 15	CONTROL-MS N = 20	CONTROL-CRYO N = 15	*p* (ANOVA)	*p* (CRYO-MS/CONTROL-MS)	*p* (CRYO-MS/CONTROL-CRYO)	*p* (CONTROL-MS/CONTROL-CRYO)
WBC (10^9^/L)	5.01 ± 1.16	4.96 ± 1.27	5.25 ± 1.40	0.786			
RBC (10^12^/L)	4.23 ± 0.61	4.30 ± 0.67	4.80 ± 0.27	0.013	0.939	0.021	0.048
HGB (g/dL)	12.31 ± 1.07	12.50 ± 1.51	13.39 ± 0.80	0.036	0.899	0.045	0.116
HCT (%)	35.97 ± 5.65	35.90 ± 5.88	40.54 ± 2.50	0.007	0.999	0.043	0.039
PLT (10^9^/L)	240.60 ± 80.97	252.70 ± 78.39	275.67 ± 35.49	0.377			
MCV (fl)	85.00 ± 5.37	83.65 ± 5.58	84.40 ± 4.03	0.738			
MCH (pg)	29.54 ± 4.03	29.42 ± 3.87	27.94 ± 1.61	0.355			
MCHC (g/dL)	34.19 ± 5.24	35.14 ± 3.37	33.05 ± 0.65	0.245			
IE0.30	0.03 ± 0.02	0.03 ± 0.02	0.05 ± 0.02	0.017	0.941	0.063	0.029
IE0.58	0.05 ± 0.01	0.05 ± 0.01	0.15 ± 0.20	0.017	0.983	0.051	0.034
IE1.13	0.11 ± 0.02	0.11 ± 0.02	0.15 ± 0.03	<0.001	0.797	<0.001	<0.001
IE2.19	0.20 ± 0.03	0.20 ± 0.02	0.21 ± 0.05	0.557			
IE4.24	0.30 ± 0.03	0.30 ± 0.03	0.26 ± 0.07	0.016	0.999	0.038	0.039
IE8.24	0.38 ± 0.04	0.38 ± 0.03	0.30 ± 0.09	<0.001	0.996	<0.001	<0.001
IE15.98	0.45 ± 0.04	0.45 ± 0.04	0.36 ± 0.09	<0.001	0.996	<0.001	<0.001
IE31.03	0.50 ± 0.04	0.50 ± 0.04	0.40 ± 0.10	<0.001	0.999	<0.001	<0.001
IE60.30	0.53 ± 0.03	0.52 ± 0.05	0.42 ± 0.09	<0.001	0.953	<0.001	<0.001
AI (%)	62.25 ± 5.65	64.27 ± 9.10	58.28 ± 8.04	0.093			
AMP (au)	19.89 ± 2.26	18.16 ± 4.57	21.62 ± 4.06	0.039	0.442	0.443	0.046
T1/2 (s)	2.26 ± 0.62	2.17 ± 0.75	2.81 ± 1.03	0.059			
Total protein (g/L)	70.17 ± 3.98	70.29 ± 2.79	73.56 ± 3.68	0.012	0.995	0.026	0.033
alb. (g/L)	44.75 ± 2.77	44.46 ± 2.60	45.85 ± 3.49	0.373			
a-1 g (g/L)	2.60 ± 0.38	2.61 ± 0.50	2.67 ± 0.33	0.871			
a-2 g (g/L)	6.29 ± 0.91	6.39 ± 0.81	5.92 ± 0.57	0.212			
b-1 g (g/L)	4.03 ± 0.60	4.19 ± 0.45	4.43 ± 0.60	0.140			
b-2 g (g/L)	3.12 ± 0.61	3.21 ± 0.67	3.19 ± 0.34	0.892			
g g (g/L)	9.41 ± 2.22	9.47 ± 2.29	11.53 ± 2.37	0.019	0.997	0.039	0.046
a/g	1.80 ± 0.29	1.76 ± 0.33	1.67 ± 0.22	0.460			
fibr. (g/L)	2.78 ± 1.30	3.18 ± 1.28	2.52 ± 0.59	0.231			

**Table 3 jcm-10-02833-t003:** ANOVA results: final parameters (after WBC) and CONTROL-MS.

Parameter	CRYO-MS N = 15	CONTROL-MS N = 20	CONTROL-CRYO N = 15	*p* (ANOVA)	*p* (CRYO-MS/CONTROL-MS)	*p* (CRYO-MS/CONTROL-CRYO)	*p* (CONTROL-MS/CONTROL-CRYO)
WBC (10^9^/L)	4.71 ± 1.01	4.96 ± 1.27	5.31 ± 1.35	0.410			
RBC (10^12^/L)	4.11 ± 0.51	4.30 ± 0.67	4.57 ± 0.28	0.112			
HGB (g/dL)	12.19 ± 1.20	12.50 ± 1.51	12.88 ± 0.73	0.310			
HCT (%)	34.70 ± 4.84	35.90 ± 5.88	38.35 ± 2.20	0.068			
PLT (10^9^/L)	238.13 ± 79.14	252.70 ± 78.39	271.73 ± 48.65	0.437			
MCV (fl)	84.27 ± 5.39	83.65 ± 5.58	84.00 ± 4.16	0.939			
MCH (pg)	30.04 ± 4.43	29.42 ± 3.87	28.25 ± 1.60	0.377			
MCHC (g/dL)	35.58 ± 4.46	35.14 ± 3.37	33.60 ± 0.42	0.832			
IE0.30	0.04 ± 0.02	0.03 ± 0.02	0.02 ± 0.01	0.045	0.470	0.035	0.354
IE0.58	0.06 ± 0.02	0.05 ± 0.01	0.07 ± 0.01	<0.001	0.193	0.029	<0.001
IE1.13	0.11 ± 0.02	0.11 ± 0.02	0.15 ± 0.02	<0.001	0.445	<0.001	<0.001
IE2.19	0.20 ± 0.02	0.20 ± 0.02	0.24 ± 0.02	<0.001	0.638	<0.001	<0.001
IE4.24	0.30 ± 0.02	0.30 ± 0.03	0.33 ± 0.02	0.001	0.958	0.009	0.004
IE8.24	0.38 ± 0.03	0.38 ± 0.03	0.40 ± 0.03	0.211			
IE15.98	0.45 ± 0.03	0.45 ± 0.04	0.47 ± 0.02	0.124			
IE31.03	0.50 ± 0.03	0.50 ± 0.04	0.52 ± 0.02	0.089			
IE60.30	0.54 ± 0.03	0.52 ± 0.05	0.56 ± 0.02	0.017	0.482	0.171	0.012
AI (%)	60.67 ± 6.33	64.27 ± 9.10	58.96 ± 7.51	0.238			
AMP (au)	21.05 ± 3.07	18.16 ± 4.57	21.91 ± 2.07	0.007	0.075	0.783	0.015
T1/2 (s)	2.50 ± 0.75	2.17 ± 0.75	2.75 ± 1.11	0.147			
Total protein (g/L)	69.64 ± 3.13	70.29 ± 2.79	71.73 ± 2.53	0.125			
alb. (g/L)	44.70 ± 2.39	44.46 ± 2.60	44.19 ± 3.13	0.878			
a-1 g (g/L)	2.53 ± 0.43	2.61 ± 0.50	2.65 ± 0.40	0.739			
a-2 g (g/L)	6.11 ± 0.65	6.39 ± 0.81	5.98 ± 0.76	0.270			
b-1 g (g/L)	4.01 ± 0.58	4.19 ± 0.45	4.55 ± 0.67	0.032	0.646	0.028	0.189
b-2 g (g/L)	2.99 ± 0.55	3.21 ± 0.67	3.20 ± 0.35	0.447			
g g (g/L)	9.35 ± 1.92	9.47 ± 2.29	11.19 ± 2.46	0.046	0.989	0.043	0.099
a/g	1.82 ± 0.26	1.76 ± 0.33	1.63 ± 0.25	0.170			
fibr. (g/L)	4.01 ± 1.70	3.18 ± 1.28	3.38 ± 1.17	0.212			

**Table 4 jcm-10-02833-t004:** Assessment of therapeutic intervention for CRYO-MS.

Parameter	CRYO-MS N = 15		*p* (t-Student/Wilcoxon)
	Before	After	
WBC (10^9^/L)	5.01 ± 1.16	4.71 ± 1.01	0.330
RBC (10^12^/L)	4.23 ± 0.61	4.11 ± 0.51	0.547
HGB (g/dL)	12.31 ± 1.07	12.19 ± 1.20	0.506
HCT (%)	35.97 ± 5.65	34.70 ± 4.84	0.482
PLT (10^9^/L)	240.60 ± 80.97	238.13 ± 79.14	0.649
MCV (fl)	85.00 ± 5.37	84.27 ± 5.39	0.308
MCH (pg)	29.54 ± 4.03	30.04 ± 4.43	0.532
MCHC (g/dL)	34.19 ± 5.24	35.58 ± 4.46	0.504
IE0.30	0.03 ± 0.02	0.04 ± 0.02	0.444
IE0.58	0.05 ± 0.01	0.06 ± 0.02	0.615
IE1.13	0.11 ± 0.02	0.11 ± 0.02	0.677
IE2.19	0.20 ± 0.03	0.20 ± 0.02	0.659
IE4.24	0.30 ± 0.03	0.30 ± 0.02	0.812
IE8.24	0.38 ± 0.04	0.38 ± 0.03	0.820
IE15.98	0.45 ± 0.04	0.45 ± 0.03	0.670
IE31.03	0.50 ± 0.04	0.50 ± 0.03	0.683
IE60.30	0.53 ± 0.03	0.54 ± 0.03	0.504
AI (%)	62.25 ± 5.65	60.67 ± 6.33	0.386
AMP (au)	19.89 ± 2.26	21.05 ± 3.07	0.092
T1/2 (s)	2.26 ± 0.62	2.50 ± 0.75	0.875
Total protein (g/L)	70.17 ± 3.98	69.64 ± 3.13	0.509
alb. (g/L)	44.75 ± 2.77	44.70 ± 2.39	0.930
a-1 g (g/L)	2.60 ± 0.38	2.53 ± 0.43	0.394
a-2 g (g/L)	6.29 ± 0.91	6.11 ± 0.65	0.232
b-1 g (g/L)	4.03 ± 0.60	4.01 ± 0.58	0.849
b-2 g (g/L)	3.12 ± 0.61	2.99 ± 0.55	0.126
g g (g/L)	9.41 ± 2.22	9.35 ± 1.92	0.699
a/g	1.80 ± 0.29	1.82 ± 0.26	0.500
fibr. (g/L)	2.78 ± 1.30	4.01 ± 1.70	0.053

**Table 5 jcm-10-02833-t005:** Assessment of therapeutic intervention for CONTROL-CRYO.

Parameter	CONTROL-CRYO N = 15		*p* (t-Student/Wilcoxon)
	Before	After	
WBC (10^9^/L)	5.25 ± 1.40	5.31 ± 1.35	0.859
RBC (10^12^/L)	4.80 ± 0.27	4.57 ± 0.28	0.007
HGB (g/dL)	13.39 ± 0.80	12.88 ± 0.73	0.014
HCT (%)	40.54 ± 2.50	38.35 ± 2.20	0.003
PLT (10^9^/L)	275.67 ± 35.49	271.73 ± 48.65	0.699
MCV (fl)	84.40 ± 4.03	84.00 ± 4.16	0.028
MCH (pg)	27.94 ± 1.61	28.25 ± 1.60	0.090
MCHC (g/dL)	33.05 ± 0.65	33.60 ± 0.42	0.013
IE0.30	0.05 ± 0.02	0.02 ± 0.01	0.001
IE0.58	0.15 ± 0.20	0.07 ± 0.01	0.161
IE1.13	0.15 ± 0.03	0.15 ± 0.02	0.611
IE2.19	0.21 ± 0.05	0.24 ± 0.02	0.011
IE4.24	0.26 ± 0.07	0.33 ± 0.02	<0.001
IE8.24	0.30 ± 0.09	0.40 ± 0.03	<0.001
IE15.98	0.36 ± 0.09	0.47 ± 0.02	<0.001
IE31.03	0.40 ± 0.10	0.52 ± 0.02	<0.001
IE60.30	0.42 ± 0.09	0.56 ± 0.02	<0.0001
AI (%)	58.28 ± 8.04	58.96 ± 7.51	0.688
AMP (au)	21.62 ± 4.06	21.91 ± 2.07	0.742
T1/2 (s)	2.81 ± 1.03	2.75 ± 1.11	0.799
Total protein (g/L)	73.56 ± 3.68	71.73 ± 2.53	0.039
alb. (g/L)	45.85 ± 3.49	44.19 ± 3.13	0.009
a-1 g (g/L)	2.67 ± 0.33	2.65 ± 0.40	0.762
a-2 g (g/L)	5.92 ± 0.57	5.98 ± 0.76	0.708
b-1 g (g/L)	4.43 ± 0.60	4.55 ± 0.67	0.136
b-2 g (g/L)	3.19 ± 0.34	3.20 ± 0.35	0.938
g g (g/L)	11.53 ± 2.37	11.19 ± 2.46	0.373
a/g	1.67 ± 0.22	1.63 ± 0.25	0.254
fibr. (g/L)	2.52 ± 0.59	3.38 ± 1.17	0.012

## Data Availability

Not applicable.

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
