# Peer review of "Effect of Whole-Body Cryotherapy on Morphological, Rheological and Biochemical Indices of Blood in People with Multiple Sclerosis"

_jcm, 2021, doi:10.3390/jcm10132833_

Round 1
Reviewer 1 Report
First of all, I would like to congratulate the authors for such as an interesting work.
Still, I think there are some points that should be improved to make the study more understandable to non-specialist readers and also to improve the formal characteristics of the article.
Abstract: the type of study is not specified. From the context I understand that it is a prospective controlled study with 3 arms, but it should be specified. The "study group" should be replaced to "experimental group" (line 12). There are some acronyms that have not been referenciated (RBC, HGB, HCT, AMP in line 25)
The term "average" is not correct, must be change to "mean" age in line 12
Keywords are missing.
Introduction: In the definition of multiple sclerosis is not necessary to give the name in latin, the acronym is enough.
When talking about the possible enviromental factors that causes MS, sentences in lines 43 to 46 are redundant and the references are not very recent (might be updated).
In lines 61, 62 the acronyms ACTH and ESR are missing.
In the following paragraphs, the references are written as "Zagrobelny i
wsp.1996" must be changed to "et al" after the name of the first author (lines 64 to 72).
Material and methods: the authors talk about "studies", but it should be a study with 3 arms, and it's important to understand each experimental condition that will be assessed and the purpose to each one to compare them and understand the results.
Inclusion criteria (without "Key"): the diagnosis of MS must be referenciated following reviewed McDonald criteria, better than the ICD10. The contraindications of the WBC must be described, or maybe included as exclussion criteria.
An important point that should be specified is if the participants signed a informed consent before iniciating the study (lines 103-105)
LIne 106: characteristics of the respondents must be changed to "characteristics of the sample/participants" because respondents is referred to the participants of a survey.
Line 107: study group must be changed to "experimental group"
Line 110: first control group, should be "control group 1". It's important to understand the purpose of this group. Was there any other intervention different form the WBC? If there is no intervention to compare, what are the parameters that the authors expect?
Line 115: the second control group shoud be "control group 2", and they were healthy controls I guess. I don't understand the purpose of the WCT if they don't have any health condition. If it's only to describe the changes in the blood parameters after the intervention in healthy individuals, I guess that's already described in the literature. Was it to compare the changes in blood parameters between healthy controls and people with MS because your intervention is different from what's been already described?
I think that the description of the intervention of the experimental group (WCT) in line 208, should be before the blood indices analysis, in line 126.
In line 166, the acronym of EI is missing
Results:
It would be necessary to include some tables to understand better the results: at least, one for the demographic characteristics of the sample, one for the pre-post results of each group, and some specific figure of the most relevant changes inter and intra groups. This would substitute a part of the written results and make more friendly the article for the readers.
Discussion:
Line 292, change Rasova i wsp. to Rasova et al. Line 375, 377, 433, 434, 435, 438, 450, 455 have the same mistake
From the second till the fifth paragraph the authors explain different concepts about rheology, erithrocytes deformability or aggregation that would fit better in the introduction, as they are defining and introducing the basis for the following discussion.
Conclussions:
After reading the article my question is: if there were no significant effect after the WCT in MS patients, why is this treatment effective? Is there is no effect, it can't be effective... This should be properly explained or corrected.
Author Response
First of all, I would like to congratulate the authors for such as an interesting work.
Still, I think there are some points that should be improved to make the study more understandable to non-specialist readers and also to improve the formal characteristics of the article.
Thank you for your praise and comments. I present my answers below and marked on the manuscript.
Abstract: the type of study is not specified. From the context I understand that it is a prospective controlled study with 3 arms, but it should be specified.
The type of study is already specified in the abstract and at the beginning of the chapter 2.1.
The "study group" should be replaced to "experimental group" (line 12). There are some acronyms that have not been referenciated (RBC, HGB, HCT, AMP in line 25)
The term "average" is not correct, must be change to "mean" age in line 12
Changed as per comments.
Keywords are missing.
Keywords: whole-body cryotherapy, multiple sclerosis, blood rheology, blood morphologyIntroduction: In the definition of multiple sclerosis is not necessary to give the name in latin, the acronym is enough.
When talking about the possible enviromental factors that causes MS, sentences in lines 43 to 46 are redundant and the references are not very recent (might be updated).
Changed to: “Environmental factors have been shown to be equally important in controlling MS prevalence and distribution, as well as some aspects of its clinical course and disability progression (Ebers, 2008; Albatineh et al., 2020). Particulate matter pollution is one of many environmental factors associated with MS risk and MS relapse (Calde-ron-Garciduenas et al., 2016). Other MS environmental associated factors include; smoking, passive or prenatal smoking, deficient sun-exposure or vitamin-D deficiency, viral infections, obesity, poor lifestyle and dietary habits, and gut microbiota (Belbasis et al., 2015; Thompson et al., 2018).”
In lines 61, 62 the acronyms ACTH and ESR are missing.
In the following paragraphs, the references are written as "Zagrobelny i
wsp.1996" must be changed to "et al" after the name of the first author (lines 64 to 72).
Material and methods: the authors talk about "studies", but it should be a study with 3 arms, and it's important to understand each experimental condition that will be assessed and the purpose to each one to compare them and understand the results.
Changed as per comments.
Inclusion criteria (without "Key"): the diagnosis of MS must be referenciated following reviewed McDonald criteria, better than the ICD10. The contraindications of the WBC must be described, or maybe included as exclussion criteria.
Changed to:
“diagnosis: multiple sclerosis – ICD10:G35 (according to McDonald crite-ria) (groups with MS)” and
“no contraindications to whole-body cryotherapy (Pregnancy, severe Hy-pertension (BP> 180/100), acute or recent myocardial infarction, unstable angina pectoris, arrhythmia, symptomatic cardiovascular disease, cardiac pacemaker, peripheral arterial occlusive disease, venous thrombosis, acute or recent cerebrovascular accident, uncontrolled seizures, Ray-naud’s Syndrome, fever, tumor disease, symptomatic lung disorders, bleeding disorders, severe anemia, infection, cold allergy, acute kidney and urinary tract diseases)”
An important point that should be specified is if the participants signed a informed consent before iniciating the study (lines 103-105)
Changed to:
“80 people applied, and 50 women were finally selected to participate in the re-search program. Before joining the study, each of the volunteers read the information for the patient and, in case of doubts, could ask questions. Then they gave their written consent to participate in the study.”
LIne 106: characteristics of the respondents must be changed to "characteristics of the sample/participants" because respondents is referred to the participants of a survey.
Line 107: study group must be changed to "experimental group"
Line 110: first control group, should be "control group 1". It's important to understand the purpose of this group. Was there any other intervention different form the WBC?
Changed as per comments.
Line 115: the second control group shoud be "control group 2", and they were healthy controls I guess. I don't understand the purpose of the WCT if they don't have any health condition. If it's only to describe the changes in the blood parameters after the intervention in healthy individuals, I guess that's already described in the literature. Was it to compare the changes in blood parameters between healthy controls and people with MS because your intervention is different from what's been already described?
There is still little research on blood rheology in the literature. In the study, we wanted to compare the changes in blood parameters between healthy controls and people with MS
I think that the description of the intervention of the experimental group (WCT) in line 208, should be before the blood indices analysis, in line 126.
In line 166, the acronym of EI is missing
Changed as per comments.
Results:
It would be necessary to include some tables to understand better the results: at least, one for the demographic characteristics of the sample, one for the pre-post results of each group, and some specific figure of the most relevant changes inter and intra groups. This would substitute a part of the written results and make more friendly the article for the readers.
The tables were from the beginning, but probably invisible to reviewers. Now added to the main part.
Discussion:
Line 292, change Rasova i wsp. to Rasova et al. Line 375, 377, 433, 434, 435, 438, 450, 455 have the same mistake
Changed as per comments.
From the second till the fifth paragraph the authors explain different concepts about rheology, erithrocytes deformability or aggregation that would fit better in the introduction, as they are defining and introducing the basis for the following discussion.
I think porting this content would make the introduction too long and extensive.
Conclussions:
After reading the article my question is: if there were no significant effect after the WCT in MS patients, why is this treatment effective? Is there is no effect, it can't be effective... This should be properly explained or corrected.
Changed as per comments.
Changes in blood rheology are not responsible for the effectiveness of treatment.

Reviewer 2 Report
Abstract:
- Please add the number of control persons in the first control group.
- In the abstract, please give full name rather than abbrevations for :RBC, HGB, HCT, and AMP
For all abbrevations, please define when it is used first time after the abstract, and later stick to the abbrevations.
Introduction:
All references are old references. Please update the introduction with newer updated information.
Material and method.
2.1.3. Characteristics of the respondents
Please specify more in details: The main criterion for qualifying for the CONTROL-MS 114 group was the organizational inability to participate in WBC procedures. As suggested below I suggest that all available data is given in a supplementary table which allows a deeper study of details for interested readers, and for further explorations.
2.2.2. Description of the intervention
Please describe the intervention more in details.
Whole body cryotherapy (WBC) was carried at the Malopolska Cryotherapy Reha210 bilitation Center in Krakow. Parameters in the cryochamber were as following:
Atrium temperature: -60 ° C
Chamber temperature: -120 ° C
– Describe: what is atrium and what is chamber ? Could a picture be added?
3 Results
In general, it is very difficult to follow the results as there is no table that gives an overview. All results are in the text. Please present a complete table of mean results to give an overview over the results.
Apply also an exploratory multivariate approach. I would suggest Principal Component Analysis (PCA). There is a possibility that different individuals within the groups may respond differently, and that is valuable information. PCA would reveal if the data within each group is homogenous or not. If there are individual variation, it could be explored more in details. For example, if some, but not all MS patients have a beneficial effect of the treatment, one could go further to explore the reason for it. For example, one could expect that the degree of inflammatory pattern could have an implication on the effect of the treatment. It is also important to be aware that there is a debate on whether inflammation in MS is a primary aspect of the disease, or not. When updating the literature in the introduction section this could be relevant point to mention.
In addition to presenting a table of all mean values in the publication, all data should be made available in supplementary table to enable further explorations. It would be useful to include as other relevant observations from the blood sample as supplemental information. All available data should be included as it may shade useful light on the results.
What are the observations in the data on oxidative damage that leads to this sentence: "It was found that although all MS patients show an increase in biomarkers of oxidative damage...". Please specify more details.
In general, do not repeat results in the discussion section. Discuss the implication of the results without repeating the results as in the result section.
4 Discussion
Add newer updated references where appropriate throughout the whole manuscript.
Change the discussion according to updated information, for example in the discussion of fibrinogen.
Correct the sentence: "Rheology is a science that studies the flow of blood through blood vessels". Rheology is wide, far beyond analysis of blood samples.
In the paragraph: "The phenomenon of erythrocyte deformability ....". Describe this in the context of the observed parameters in the present study, thus discuss this in light of the observed elongation and aggregation index.
Please specify and give more details on markers of oxidative damage. The text says: “It was found that although all MS patients show an increase in biomarkers of oxidative damage, there seems to be no correlation between the degree of height and the severity of the disease”. How does this sentence refer to the observed data.
Please change the sentence: The conducted own research is probably the first … Say instead: “To the best of our knowledge, the present study is the first…”
Author Response
Abstract:
Please add the number of control persons in the first control group.
In the abstract, please give full name rather than abbrevations for :RBC, HGB, HCT, and AMP
For all abbrevations, please define when it is used first time after the abstract, and later stick to the abbrevations.
Changed as per comments.
Introduction:
All references are old references. Please update the introduction with newer updated information.
More up-to-date sources added.
“Environmental factors have been shown to be equally important in controlling MS prevalence and distribution, as well as some aspects of its clinical course and disability progression (Ebers, 2008; Albatineh et al., 2020). Particulate matter pollution is one of many environmental factors associated with MS risk and MS relapse (Calde-ron-Garciduenas et al., 2016). Other MS environmental associated factors include; smoking, passive or prenatal smoking, deficient sun-exposure or vitamin-D deficiency, viral infections, obesity, poor lifestyle and dietary habits, and gut microbiota (Belbasis et al., 2015; Thompson et al., 2018).”
Material and method.
2.1.3. Characteristics of the respondents
Please specify more in details: The main criterion for qualifying for the CONTROL-MS 114 group was the organizational inability to participate in WBC procedures. As suggested below I suggest that all available data is given in a supplementary table which allows a deeper study of details for interested readers, and for further explorations.
“The main reason for qualifying for the CONTROL-MS group was the organizational inability to participate in WBC procedures (comparison of the baseline).”
The tables were from the beginning, but probably invisible to reviewers. Now added to the main part.
2.2.2. Description of the intervention
Please describe the intervention more in details.
Whole body cryotherapy (WBC) was carried at the Malopolska Cryotherapy Reha210 bilitation Center in Krakow. Parameters in the cryochamber were as following:
Atrium temperature: -60 ° C
Chamber temperature: -120 ° C
– Describe: what is atrium and what is chamber ? Could a picture be added?
Chamber diagram has been added. (please find the fig in attachment)
General diagram of a cryochamber (according to CREATOR Sp. z o.o)
3 Results
In general, it is very difficult to follow the results as there is no table that gives an overview. All results are in the text. Please present a complete table of mean results to give an overview over the results.
In addition to presenting a table of all mean values in the publication, all data should be made available in supplementary table to enable further explorations. It would be useful to include as other relevant observations from the blood sample as supplemental information. All available data should be included as it may shade useful light on the results.
The tables were from the beginning, but probably invisible to reviewers. Now added to the main part.
What are the observations in the data on oxidative damage that leads to this sentence: "It was found that although all MS patients show an increase in biomarkers of oxidative damage...". Please specify more details.
We describe the pro-oxidative-antioxidant balance in another work (in the process of publication).
In general, do not repeat results in the discussion section. Discuss the implication of the results without repeating the results as in the result section.
In the discussion, I am merely referring to our results for the sake of clarite
4 Discussion
Add newer updated references where appropriate throughout the whole manuscript.
Change the discussion according to updated information, for example in the discussion of fibrinogen.
Correct the sentence: "Rheology is a science that studies the flow of blood through blood vessels". Rheology is wide, far beyond analysis of blood samples.
In the paragraph: "The phenomenon of erythrocyte deformability ....". Describe this in the context of the observed parameters in the present study, thus discuss this in light of the observed elongation and aggregation index.
Please specify and give more details on markers of oxidative damage. The text says: “It was found that although all MS patients show an increase in biomarkers of oxidative damage, there seems to be no correlation between the degree of height and the severity of the disease”. How does this sentence refer to the observed data.
Please change the sentence: The conducted own research is probably the first … Say instead: “To the best of our knowledge, the present study is the first…”
Changed as per comments.

Round 2
Reviewer 1 Report
Thank you very much for your clear and comprehensive corrections of the manuscript.
There are still some minor mistakes that might be corrected before my definitive acceptation:
- Abstract: in line 12, the type of study can not be between brackets. It should go in the following sentence, something like "In this prospective controlled study, the experimental group consisted of..." In line 15 the size of the sample can not be between brackets, something like " The first control group consisted of 20 women with..." In lines 27,28 the names of the blood cells should go before the acronym (this last one going between brackets), like "hematocrit (HCT)..."
- Introduction: lines 71-73, again the names of the hormone before the acronym "Adrenocorticotropic Hormone (ACTH), erythrocyte sedimentation rate (ESR)"
- Material: line 94, the type of study is not clear. It is one study with 3 arms, not 3 different independent studies, if we want to compare them, isn't it? Then the sentence should be something like "This prospective controlled study with 3 arms was carried out..."
- Inclusion criteria: line 102, should be "diagnosis of MS according to Mc Donald review criteria" and then, the reference:
Thompson AJ, Banwell BL, Barkhof F, et al. Diagnosis of multiple sclerosis: 2017 revisions of the McDonald criteria. Lancet Neurol. 2018;17(2):162-173.
- I think the contraindications to the WCT (line 197-113)would be exclusion criteria, better than non having those contraindication is an inclusion criteria.
- Conclussions: (line 573-574) Avoid brackets in the sentence. It can go like this "WBC is an effective and safe form of therapy in MS patients as changes in blood rheology are not responsible for the effectiveness of treatment".
Author Response
Abstract: in line 12, the type of study can not be between brackets. It should go in the following sentence, something like "In this prospective controlled study, the experimental group consisted of..." In line 15 the size of the sample can not be between brackets, something like " The first control group consisted of 20 women with..." In lines 27,28 the names of the blood cells should go before the acronym (this last one going between brackets), like "hematocrit (HCT)..."
Changed as per comments.
Introduction: lines 71-73, again the names of the hormone before the acronym "Adrenocorticotropic Hormone (ACTH), erythrocyte sedimentation rate (ESR)"
Changed as per comments.
Material: line 94, the type of study is not clear. It is one study with 3 arms, not 3 different independent studies, if we want to compare them, isn't it? Then the sentence should be something like "This prospective controlled study with 3 arms was carried out..."
Truth, changed as per comments.
Inclusion criteria: line 102, should be "diagnosis of MS according to Mc Donald review criteria" and then, the reference: Thompson AJ, Banwell BL, Barkhof F, et al. Diagnosis of multiple sclerosis: 2017 revisions of the McDonald criteria. Lancet Neurol. 2018;17(2):162-173.
I think the contraindications to the WCT (line 197-113)would be exclusion criteria, better than non having those contraindication is an inclusion criteria.
Changed as per comments.
Conclussions: (line 573-574) Avoid brackets in the sentence. It can go like this "WBC is an effective and safe form of therapy in MS patients as changes in blood rheology are not responsible for the effectiveness of treatment".
Changed as per comments.

Reviewer 2 Report
Abstract
Line 25-27: Specify what is compared in the sentence:
Results: statistically significant differences and changes in levels of RBC …..
Introduction
Line 59, correct the sentence, there are two times deficient.
Results
There are no references to figures or tables in the result section. References to the tables should be included. The table texts should be improved to better explain what the tables present.
The figure labels says: “study 1 and study 2”, but that is not described in the method section or the result section. There are abbreviations in the figures that are not explained, such as KRIO-SM in Figure 2 and SS in Figure 5.
Multivairate explorative analysis such as Principal Component Analysis (PCA) is recommended to obtain an overview over the results and also to visualize individual differences within groups.
Discussion
There is a need to update the discussion with newer references. For example, in the first sentence it is commented that no effective therapy has yet been developed to combat the disease. A reference from 2006 can not be used as the only reference to this statement. Throughout the discussion there is a need to study and include also newer references.
Results such as p-values does not belong to the discussion part. It is more confusing that helpful. As it is now there is a result section that is too short, and then a discussion section which is a combination of results and discussion. My suggestion would be: alternative 1: to combine results and discussion into one section “results and discussion”, or alternative 2: to better distinguish the two to have all results in the result section and omit results such as p-values etc in the discussion section.
Conclusions
The study to not detect effects of WBC for MS patients, yet the conclusion state that WBC is an effective and safe therapy for MS. What is the basis for this conclusion?
Author Response
Abstract
Line 25-27: Specify what is compared in the sentence:
Results: statistically significant differences and changes in levels of RBC …..
Changed as per comments.
Introduction
Line 59, correct the sentence, there are two times deficient.
Changed as per comments.
Results
There are no references to figures or tables in the result section.
References to the tables should be included. The table texts should be improved to better explain what the tables present.
The figure labels says: “study 1 and study 2”, but that is not described in the method section or the result section. There are abbreviations in the figures that are not explained, such as KRIO-SM in Figure 2 and SS in Figure 5.
Changed as per comments.
Discussion
There is a need to update the discussion with newer references. For example, in the first sentence it is commented that no effective therapy has yet been developed to combat the disease. A reference from 2006 can not be used as the only reference to this statement. Throughout the discussion there is a need to study and include also newer references.
Results such as p-values does not belong to the discussion part. It is more confusing that helpful. As it is now there is a result section that is too short, and then a discussion section which is a combination of results and discussion. My suggestion would be: alternative 1: to combine results and discussion into one section “results and discussion”, or alternative 2: to better distinguish the two to have all results in the result section and omit results such as p-values etc in the discussion section.
Changed as per comments.
A small amount of research on this topic resulted in the citation of older items, also valuable.
Conclusions
The study to not detect effects of WBC for MS patients, yet the conclusion state that WBC is an effective and safe therapy for MS. What is the basis for this conclusion?
Changed as per comments.
